# Impact of Dairy Intake on Plasma F_2_-IsoProstane Profiles in Overweight Subjects with Hyperinsulinemia: A Randomized Crossover Trial

**DOI:** 10.3390/nu13062088

**Published:** 2021-06-18

**Authors:** Leila Khorraminezhad, Jean-François Bilodeau, Karine Greffard, Jessica Larose, Iwona Rudkowska

**Affiliations:** 1Endocrinology and Nephrology Unit, CHU de Québec-Laval University Research Center, CHUL-2705, Quebec City, QC G1V 4G2, Canada; Leila.khorraminezhad.1@ulaval.ca (L.K.); Karine.Greffard@crchudequebec.ulaval.ca (K.G.); Jessica.Larose@crchudequebec.ulaval.ca (J.L.); 2Department of Medicine, Laval University, Quebec City, QC G1V 4G2, Canada; Jean-Francois.Bilodeau@crchudequebec.ulaval.ca

**Keywords:** F_2_-isoprostane, dairy products, type 2 diabetes, hyperinsulinemia

## Abstract

F_2_-IsoProstanes (F_2_-IsoPs) are major biomarkers of oxidative stress and are associated with type 2 diabetes (T2D). Further, plasma levels of F_2_-IsoPs may be modified by dairy products. The aim is to investigate the effect of high dairy product (HD) consumption compared to an adequate dairy product (AD) consumption on the level of F_2_-IsoPs among hyperinsulinemic subjects. In this crossover study, participants were randomized in two groups: HD (≥4 servings/day), or AD (≤2 servings/day) for six weeks. Fasting blood glucose and insulin were measured. The homeostatic model assessment of insulin resistance (HOMA-IR) was calculated. Six isomers of F_2_-IsoPs were quantified by HPLC-MS/MS. Twenty-seven subjects with hyperinsulinemia (mean age; 55 ± 13 years, BMI; 31.4 ± 3.3 kg/m2) were included. Fasting glucose, insulin and HOMA-IR were unchanged after HD or AD intervention. After HD intake, the total level of F_2_-IsoPs (*p* = 0.03), 5-F_2*t*_-IsoP (*p* = 0.002), and 8-F_2*t*_-IsoP (*p* = 0.004) decreased compared to AD. The 15-F_2t_-IsoP tended to be positively correlated with fasting blood glucose (*r* = 0.39, *p* = 0.08). Generally, F_2_-IsoPs levels were higher among men compared to women regardless of the dairy intake. Overall, intake of HD decreased plasma levels of F_2_-IsoPs compared to AD without modifying glycemic parameters.

## 1. Introduction

F_2_-isoprostanes (F_2_-IsoPs) are prostaglandin-like compounds produced from the oxidation of polyunsaturated fatty acids (PUFA) like arachidonic acid (AA) [1]. F_2_-IsoPs are used as gold standard markers for the assessment of oxidative stress [2]. Isoprostanes (IsoPs) can be generated in phospholipids by free radicals and are subsequently released from cell membranes by phospholipases [3]. Generally, F_2_-IsoPs are associated with atherogenesis and inflammation. F_2_-IsoPs are believed to act through impairment of glycemic homeostasis [3], stimulation of proliferative responses in fibroblasts [4] and alterations of membrane lipids [5].

The level of plasma F_2_-IsoP has been recognized as one of the major biomarkers associated with the pathogenesis of type 2 diabetes (T2D) [4]. F_2_-IsoPs have been implicated in the impairment of beta-cell functions and have been shown to induce apoptosis through the peroxidation of membrane cholesterol [6]. Moreover, observational and clinical studies confirmed an increase in total levels of F_2_-IsoPs, specifically the 15-F_2t_-IsoP, also known as the classical 8-iso-PGF_2a_, among diabetic patients in comparison to nondiabetic [3,4,5,6] and healthy overweight adults [7]. Overall, F_2_-IsoPs are associated with major inflammatory parameters that determine the incidence of T2D among obese populations [2].

Dietary intake, specifically of dairy products, could contribute to the prevention of oxidative stress and an adequate glycemic response through micronutrient effects such as calcium and vitamin D. For instance, a randomized-parallel-group study indicated a reduction in plasma level of 15-F2t-IsoP after intake of adequate-dairy (3.5 servings/d and ≥1000 mg calcium/d) compared to low-dairy (<0.5 daily servings and ≤600 mg calcium/d) during a 12-week intervention in obese adults with metabolic syndrome [8]. However, a 6-month randomized, parallel-group intervention study demonstrated no change in the plasma level of 15-F2t-IsoP after consumption of three to five milk servings per day compared to the control group (0.5–1 serving/d) among overweight men and women [9]. Despite indirect evidence of the potential role of dairy products on oxidative stress and glycemic parameters, the association between T2D risk factors and F2-IsoPs after dairy product intake remains unknown in overweight individuals with hyperinsulinemia.

Thus, the hypothesis of this study is that the level of F_2_-IsoPs are modified after consumption of high dairy (HD (≥4 servings/day)) products compared with consumption of adequate dairy (AD (≤2 servings/day)) products. The secondary hypothesis is that variations in the levels of F_2_-IsoPs are correlated with changes in glycemic parameters among subjects with hyperinsulinemia.

## 2. Materials and Methods

### 2.1. Participants

This randomized crossover study included 27 individuals (19 men and 8 women) aged between 28 and 71 years who were assessed at the CHU de Québec-Université Laval Research Center from February 2017 to July 2018. All subjects were Caucasian men and postmenopausal women (absence of menstrual cycles for >12 months) aged >18 yrs, BMI between 25–40 kg/m2 (overweight, 25 ≤ BMI ≤ 29.9 kg/m^2^ or obesity, BMI ≥ 30 kg/m^2^), had hyperinsulinemia and selected based on fasting insulin > 90 pmol/L, fasting glucose < 7.0 mmol/L and glycated hemoglobin (HbA1c) < 6.5%. Furthermore, exclusion criteria were the same as in the previously published study [10], including high dairy consumption (≥2 servings/day), major surgery in the three months prior to study onset, smoking, incompatibility with dairy consumption (allergy, intolerance or dislike), inflammatory bowel disease or other gastrointestinal disorder influencing gastrointestinal motility or nutrient absorption, medications known to affect lipid and glucose metabolism other than those used to treat hypertension or dyslipidemia and diseases known to affect glucose metabolism.

This study was approved by the CHU de Québec-Université Laval Research Center ethics committees (permission code: 2017-3228) and was conducted according to the principles of the declaration of Helsinki. All subjects provided written informed consent. This trial protocol is registered in ClinicalTrial.gov: NCT02961179. Subjects deemed eligible through a telephone screening process were invited for a screening visit where demographic, medical and food frequency questionnaires were administered.

### 2.2. Dietary Intervention

Participants were equally randomized to HD (n = 12) or AD (n = 15). After the first intervention phase (6-week) and a 6-week washout period, subjects crossed over to the other phase (6-week). During the HD intervention period, subjects consumed ≥4 servings/day of dairy products and during the AD intervention period, they consumed ≤2 servings/day of dairy products (according to the serving sizes recommended by Canada’s Food Guide for Healthy Eating (2007) [11]). All subjects were requested to keep the same physical activity, eating habits and other aspects of lifestyle during the study. Dairy products comprised milk, yogurt, cheese, kefir and cream (≤15% fat content). Ice cream intake (1 serving equals 125 mL) was limited to three servings per week. Butter, milk substitutes and derivatives, whipped cream or cream >15% fat content, and soy desserts or plant-based beverages (almond, cashew, rice, hemp, etc.) were not accepted as dairy products in the daily serving count. Dietary information was collected at each visit by using a validated autoadministered food frequency questionnaire (FFQ) containing 91 items and 33 subquestions that were completed through a web platform linked to the Nutrition Data System for Research [12]. Dietary intake was estimated using the Canadian Nutrient File 2015 [11].

### 2.3. Anthropometric Assessments

For all participants in the study, weight (kg) and height (cm) were measured. Body weight was measured with a professional scale accurate to 0.1 kg (Health O Meter Professional, Sunbeam products, Inc., Boca Raton, FL, USA), and height was measured using a wall-mounted stadiometer with 1 mm accuracy (The Easy-Glide Bearing Stadiometer, Perspective Enterprises), with subjects in light clothing and without shoes. Body mass index (BMI) was calculated as weight (kg) divided by height (m) squared and recorded as kg/m^2^. Blood pressure was measured following a 15-min rest on a chair by using the SphygmoCor XCEL system. Body fat mass was evaluated in the fasting state and at the same time across visits using a 4-electrode bioimpedance scale (InBody 520 Body Composition Analyzer).

### 2.4. Clinical Measurements

Visits were scheduled at the beginning and at the end of each intervention period, which represented a total of four visits separated by six week intervals. Fasting blood glucose (FBG) levels were measured using hexokinase assays [13]. Blood fasting insulin levels were measured with chemiluminescence immunoassays [14]. Insulin resistance was estimated using the HOMA-IR index: HOMA-IR = [insulin (pmol/L) × glucose (mmol/L)]/135 [15]. Plasma total cholesterol (TC) and triglycerides (TG) concentrations were measured using enzymatic assays [16,17]. The HDL cholesterol (HDL-C) fraction was obtained after precipitation of very low-density lipoprotein and LDL particles in the infranatant with heparin manganese chloride [18]. LDL cholesterol (LDL-C) was calculated with the Friedewald formula [19]. HbA1c was determined using a colorimetric method after an initial separation by ion exchange chromatography [20].

### 2.5. Measurements of Plasma Level of F_2_-IsoPs

Total F_2_-IsoPs and its isomers, including 5(*RS*)-5-F_2c_-IsoP, 5-*epi*-5-F_2t_-IsoP, 5-F_2t_-IsoP, 8-F_2t_-IsoP, 15-*epi*-15-F_2t_-IsoP and 15-F_2t_-IsoP were measured in plasma from a modified previous method [21]. Briefly, 750 µL of plasma were mixed with 250 µL of water and 10 µL of deuterated internal standards (25 ng of deuterated standard in ethanol). Alkaline hydrolysis was performed by adding 985 µL of potassium hydroxide (1 M in methanol) for 1 h at 37 °C. Water (1 mL) was further added to the sample before proceeding to a solid phase extraction (SPE) with Oasis MAX 60 mg/3 cc cartridge from Waters Corporation (Mississauga, ON, Canada). Briefly, samples were loaded on SPE cartridges conditioned with methanol and water. Consecutive washes were made with sodium acetate buffer (50 mM, pH 3), n-hexane, ammonium hydroxide (2.5 mM), and finally with an acetonitrile-methanol mixture (8:2 *v*/*v*). The F_2_-IsoPs were eluted with a mix of 79.2% of acetonitrile, 19.8% methanol and 1% acetic acid and then dried under a nitrogen stream. Samples were reconstituted with 60 μL of 30% acetonitrile-0.01% acetic acid. The same procedure was used to build a calibration curve in plasma. The analysis was carried out on a Shimadzu Prominence HPLC (Columbia, MD, USA) coupled to a 3200 QTRAP^®^ MS/MS from AB Sciex (Concord, ON, Canada) configured with a Turbo V™ electrospray ionization probe operated in negative mode. Chromatographic separation of injected samples (40 µL) was achieved on a Kinetex C18 100 Å column from Phenomenex (Torrance, CA, USA) with a ternary mobile phase gradient described elsewhere [22]. Acquisition with Analyst 1.7 was performed in the multiple reactions monitoring mode and quantification was achieved by using SCIEX OS 1.7MQ software (AB Sciex).

### 2.6. Statistical Analyses

Statistical analyses were performed using IBM SPSS statistics software for macOS version 27.0.1.0 (IBM Corp. Armonk, (NY) the USA). The charts were generated by GraphPad Prism version 9 for macOS. Before performing statistical analyses, the Shapiro–Wilk test was examined to evaluate the normality of the data distribution The difference in IsoPs levels at the beginning of each phase between AD and HD was examined by an independent sample *t*-test. The generalized linear mixed model (GLMM) was performed with repeated measures (visit) and compound symmetry as a repeated covariance structure that minimizes the Akaike criterion. A collinearity test was conducted revealing collinearity between BMI and energy intake (*p* = 0.03). Therefore, the fixed effects in GLMM analysis were intervention (HD, AD), visit number (V2, V4), sequence (AD/HD, HD/AD), sex and BMI. The subject was considered as a random effect. Robust estimation was used to handle the violation of model assumptions. Multiple comparisons were adjusted using sequential Bonferroni. Bivariate and partial correlations (adjusted for sex and BMI) were applied to examine the quantitative relationship between IsoPs levels and glycemic profile (FBG, fasting blood insulin, HOMA-IR).Data are expressed as arithmetic means ± SD unless otherwise stated. A *p*-value of less than 0.05 was considered significant for all statistics, and a *p*-value of between 0.05 and 0.1 was considered a tendency.

## 3. Results

### 3.1. Descriptive Results of Rarticipants

The study flow diagram is published in the previous study (Appendix A) [10]. This crossover study included 27 subjects with hyperinsulinemia (2 h glucose post-OGTT ≥7.8 mmol/L and/or FBG ≥6.1 mmol/L) of both sexes (8 women, 19 men). The characteristics of participants are presented in Table 1. The baseline means of age, BMI, weight, height and HbA1c of all participants were 55 ± 13 years, 31.4 ± 3.3 kg/m^2^, 90.33 ± 14.77 kg, 1.69 ± 0.09 m, and 5.58 ± 0.29 %, respectively. Participants were categorized as overweight (7 subjects, 25 ≤ BMI ≤ 29.9 kg/m^2^) and obese (20 subjects) group (BMI ≥ 30 kg/m^2^). GLMM analysis indicated that after intake of HD, the level of anthropometric characteristics, glycemic parameters (FBG, fasting blood insulin, HOMA-IR) and lipid profiles (LDL, HDL, TG, Chol) were unchanged compared to AD consumption.

### 3.2. Dietary Intake

Dietary intakes from FFQ are presented in Table 2. The mean of dairy products consumption after AD compared with HD was 2.4 ± 1.2 and 5.8 ± 2.1 servings/d, respectively. The portions intake of milk (0.77 ± 0.72 compared to 2.43 ± 1.41, *p* < 0.001(AD vs. HD, respectively)), cheese (fatty-cheese and nonfatty cheese) (0.80 ± 0.80 compared to 1.94 ± 1.04, *p* < 0.001), yogurt (0.31 ± 0.42 compared to 1.49 ± 1.52, *p* < 0.001), liquid dairy products (1.21 ± 0.79 compared to 4.19 ± 1.95, *p* < 0.001), low-fat dairy (1.11 ± 0.89 compared to 3.65 ± 2.13, *p* < 0.001), and fermented dairy (1.07 ± 0.896 compared to 3.18 ± 1.59, *p* < 0.001) were higher after HD compared to AD.

Furthermore, GLMM analysis demonstrated that total energy intake (*p* = 0.01), total intake of macronutrient including carbohydrate (*p* = 0.04), protein (*p* = 0.001), fat (*p* = 0.01), and saturated fat (*p* = 0.002) were significantly higher after HD compared to AD. However, estimated consumption of AA, eicosapentaenoic acid (EPA) and docosahexaenoic acids (DHA) as primary precursors in the synthesis of IsoPs remained unchanged after HD intervention compared to AD intervention (*p* > 0.05).

### 3.3. Isomers of F_2_-IsoPs

No difference was found in the level of total F_2_-IsoPs and isomers of F_2_-IsoPs at the beginning of each phase by independent sample *t*-test (*p* > 0.05) (Appendix A). Several isomers of F_2_-IsoPs, including 5(*RS*)-5-F_2c_-IsoP, 5-*epi*-5-F_2t_-IsoP, 5-F_2t_-IsoP, 8-F_2t_-IsoP, 15-*epi*-15-F_2t_-IsoP and 15-F_2t_-IsoP were measured after HD and AD consumption (Figure 1). Using paired *t*-test, a significant lower plasma levels of both, 5-F_2t_-IsoP (-21%; *p* = 0.003) and 8-F_2t_-IsoP (−16%; *p* = 0.008) after consumption of HD compared to AD were observed (Figure 1). Furthermore, changes in the level of isoprostane during AD and HD phases were indicated in the Appendix A. There was no significant difference in the changes of IsoPs level between AD and HD phases. Due to the effect of energy intake on IsoPs levels in previous studies, we examined the correlation between energy intake and IsoPs levels in each phase; there were no correlation between energy intake and IsoPs levels in present study.

GLMM analysis that took into account BMI and sex showed that after HD intake the total levels of F_2_-IsoP (*p* = 0.03, −11%), 5-F_2t_-IsoP (*p* = 0.002, −16%) and 8-F_2t_-IsoP (*p* = 0.004, −15%) were lower in comparison to AD (Table 3; Figure 2).

Of note, there was no effect of age and energy intake on isoprostane levels in GLMM analyses; therefore, sex and BMI were considered as fixed effect parameters. Most F_2_-IsoPs isomers were affected by the sex of the participant except for the 15-F_2t_-IsoP. Indeed, a higher level of F_2_-IsoPs isomers in men than women was observed in general (Figure 2).

Moreover, no carry-over effects were found in the study, since the sequence of the treatment remained nonsignificant (Table 3). Of note, no correlation was found between isomers of F_2_-IsoPs and glycemic profile (FBG, fasting blood inulin, HOMA-IR) after AD and HD intervention. Furthermore, after adjusting for age, sex and BMI in AD intervention, the 15-F_2t_-IsoP tended to be positively correlated with FBG (*p* = 0.08; *r* = 0.39; Appendix A).

## 4. Discussion

The present study demonstrates for the first time that the consumption of HD decreases the total level of F_2_-IsoP, 5-F_2t_-IsoP, and 8-F_2t_-IsoP compared to consumption of AD. Specifically, the 5-F_2t_-IsoP and 8-F_2t_-IsoP were reduced by HD intake. Most previous studies investigated only the 15-F_2t_-IsoP. Moreover, 15-F_2t_-IsoP was positively correlated with FBG after AD consumption. These results also demonstrate the need to assess several isomers of F_2_-IsoP to obtain an overview of oxidative effects. The specific isomers affected can differ with pathologies. The previous study showed that 5-F_2t_-IsoP/5-epi-F_2t_-IsoP were associated with hypertension during pregnancy and proteinuria [23], while 15-*epi*-15-F_2*t*_-IsoP was associated with diabetes in pregnancy [24]. Moreover, plasma levels of F_2_-IsoP were lower among women compared to men after HD consumption compared to AD.

Generally, the level of F_2_-IsoPs was lower after consumption of HD compared to AD, especially the total level of F_2_-IsoP, 5-F_2t_-IsoP, 8-F_2t_-IsoP, and 15-F_2t_-IsoP. Similarly, the plasma levels of 15-F_2t_-IsoP and tumor necrosis factor-alpha (TNF-alpha) were reduced after the intake of 1200-1400 mg calcium/d (three servings/d dairy products and 1050 mg calcium/d) compared with 400-500 mg calcium/d (soy-supplementation) during a four-week period in obese or overweight subjects [5]. However, no difference in the plasma level of F_2_-IsoPs was observed after consumption of low-fat dairy (400 mL milk/d and 200 g yogurt/d), fermented dairy (85 g cheddar cheese/d and 600 g full-cream yogurt/d) and nonfermented dairy (30 g butter/d and 70 mL cream/d) among overweight and obese subjects during a three-week period [7]. Parallel results were reported among young athletes, who had no differences in the plasma total level of F_2_-IsoPs between three interventions, including consumption of 500 mL/day dough (liquid yogurt), nonalcoholic beer, or chocolate milk after a 2-h intake of beverages [25]. In sum, the quantity and type of dairy product consume could modify the level of F_2_-IsoPs to a certain extent and specifically 5-F_2t_-IsoP, 8-F_2t_-IsoP.

As expected, total energy intake was increased after HD compared to AD intervention, which included an increase in total protein, total carbohydrate and total fat, specifically SFA. Consumption of total energy and macronutrients can be associated with the plasma level of IsoPs. For example, intake of energy, total fat and SFA among women smokers was higher than in women nonsmokers, causing elevation of urinary total F_2_-IsoP [26]. Furthermore, no difference was indicated in the plasma level of 8-iso-PGF_2α_ after consumption of a milkshake, with 59% of total energy as fat (including refined olive oil, palm oil, olive oil plus 4 g n-3FA), among obese and overweight adults [27]. Moreover, the Framingham cohort study showed that urinary inflammation and oxidative stress (total isoprostane) parameters were inversely associated with total protein and plant protein [28]. Further, a cross-sectional study on healthy youth with weight stable for three months and an isocaloric diet (50% carbohydrate, 30% fat, 20% protein), reported no association between urinary 15-F_2t_-IsoP and basal carbohydrate and lipid oxidation rate [29]. Contrary to the current study, no differences were demonstrated in consumption of total energy, carbohydrate, protein and fat between different quartiles of F_2_-isoP in a cross-sectional study among postmenopausal women [30]. In sum, IsoPs levels can be associated positively with the intake of fat, SFA, and total energy as demonstrated in previous studies while protein intake can reduce the urinary or plasma levels of isoPs and oxidative stress.

The results indicate a positive correlation between 15-F_2t_-IsoP and FBG after AD intervention. Similarly, our research group previously demonstrated that higher plasma 15-F_2*t*_-IsoP (up to 67%) in diabetic than nondiabetic pregnant women without any difference in the level of PUFA (n-6) and AA, while the plasma level of EPA and DHA were lower in diabetic than normoglycemia pregnant women [24]. Moreover, the plasma level of 15-F_2*t*_-IsoP, was higher in prediabetic patients compared to individuals with normal glucose tolerance [31]. The plasma level of 15-F_2*t*_-IsoP was even higher among T2D patients than prediabetic [32] patients, with a negative correlation between 8-iso-PGF_2α_ and HbA_1C_ [31]. Further, a recent cohort study among the German population indicated that urinary 15-F_2*t*_-IsoP levels were positively associated with the development of T2D (increase of HbA_1C_) among elderly participants (>60 years), especially of individuals between 65 and 75 years. Similarly, the urinary levels of 15-F_2*t*_-IsoP and 8-Oxo-2′-deoxyguanosine (as a biomarker of oxidative DNA damage) increased (up to 65%) and among T2D patients compared to nondiabetic controls; therefore, 15-F_2*t*_-IsoP was considered as a biomarker to early prediction of T2D [33]. The current results demonstrate the role of F_2_-IsoP and its isomers such as 15-F_2t_-IsoP, 5-F_2t_-IsoP and 5(*RS*)-5-F_2*c*_-IsoP may be involved in the development of T2D.

The current study showed that the level of F_2_-IsoP was lower among women compared to men after consumption of HD compared to AD. Similar results were found in an international multicenter (Italy, Poland, and Sweden) study, with lower (estimated −8%) urinary level of 15-F_2*t*_-IsoP among women than men, which may be associated with the lower lean body mass among women compared to men [34]. Moreover, urinary 15-F_2*t*_-IsoP, as the major F_2_-IsoP generated through lipid peroxidation, was higher (52%) among overweight/obese men compared to women. In addition, the baseline level of 5-F_2*t*_-IsoP was higher among women than men while, after adjusting for sex and BMI, urinary levels of 5-F_2*t*_-IsoP were reduced by 20% and 27% at 12 and 18 months after intake of a 25% calorie restriction [35]. Therefore, lower levels of F_2_-IsoP in women than in men could be expected [36] among hyperinsulinemic women compared to men.

There were strengths and limitations in the present study. Major strengths included a controlled and randomized design, inclusion of participants from both sexes, and a six-week intervention for the laboratory parameters and dietary intake throughout the study. Moreover, the short duration of the intervention (six weeks) was considered a limitation of the present study; perhaps longer studies are needed to make changes in T2D risk factors. Furthermore, FFQs under and overestimate intake of food and it are considered a low accuracy method to measure absolute nutrient values. Further, the information for the availability of dietary components such as fatty acids is not sufficient. However, FFQs can assess usual and longer-term dietary intake. In addition, future studies should investigate whether specific types of dairy intake (e.g., milk, yogurt and cheese) can modify glycemic parameters (HOMA-IR, fasting insulin, FBG and HbA1c) among prediabetic and diabetic patients. Examination of molecular and metabolic pathways should also be done to understand the modifications of antioxidant and inflammatory responses after consumption of dairy products. Moreover, participants consumed more energy after HD compared to AD, therefore, design of an isocaloric diet to examine the effect of dairy products on oxidative stress in future studies suggest is suggested.

Overall, longer studies should clarify whether the optimal number of servings per day, and the type of dairy, to prevent oxidative effects among overweight and obese subjects with hyperinsulinemia.

## 5. Conclusions

The study clearly shows that the consumption of four servings or more of dairy per day compared to two servings or less reduced the total level of F_2_-IsoPs and specific isomers (5-F_2t_-IsoP, 8-F_2t_-IsoP) that could serve as markers of hyperinsulinemia in the future. This phenomenon is independent of FBG, fasting insulin and HOMA-IR, that remained unchanged between AD and HD. A reduction of oxidative damage following HD may reduce or slow down T2D development. Therefore, the link between oxidative stress and insulin resistance needs to be further investigated.

## Figures and Tables

**Figure 1 nutrients-13-02088-f001:**
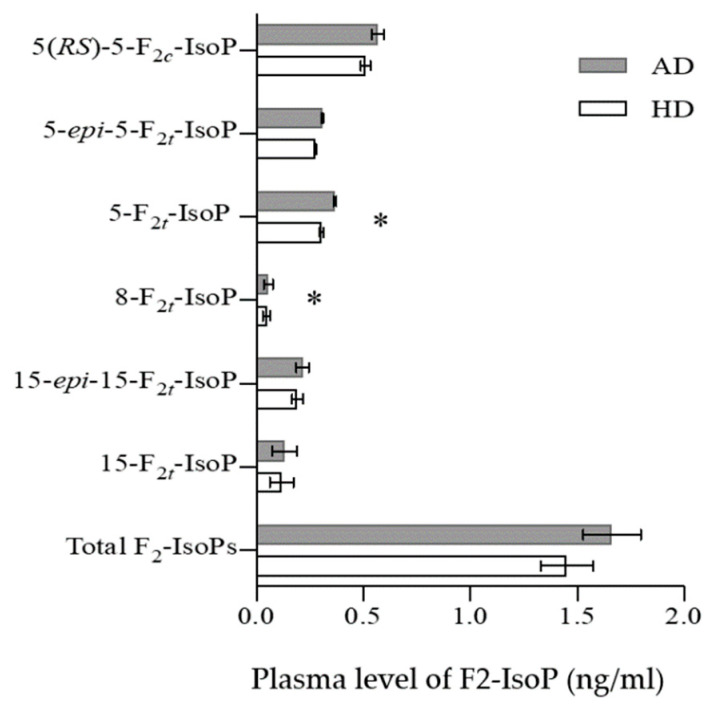
Plasma levels of F_2_-IsoProstane (F_2_-IsoPs) (ng/mL) in response to adequate-dairy (AD) and high-dairy diet (HD). Data are means ± SD (n = 27). * *p* < 0.05 significant.

**Figure 2 nutrients-13-02088-f002:**
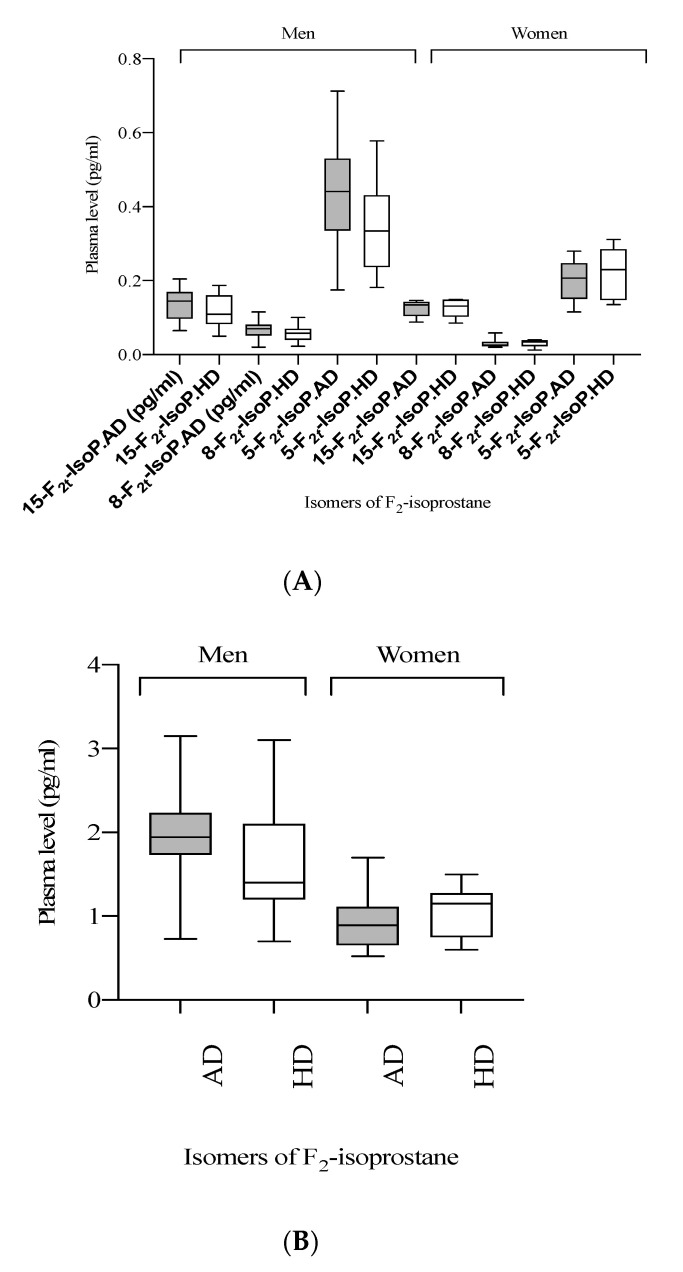
Boxplot of estimated marginal means of difference in the plasma level of F2-IsoP (ng/mL) according to sex after high-dairy (HD) and adequate-dairy (AD) diet interventions. (**A**) levels of specific F_2_-IsoP isomers and (**B**) sums of all F_2_-IsoPs. Generalized linear mixed model (GLMM) analyses with two covariates (BMI and sex), repeated statements (visit).

**Table 1 nutrients-13-02088-t001:** Coefficient estimates from GLMM analyses for main characteristic of subjects at the end of each phase following adequate dairy product intake (AD) and high dairy product intake (HD).

Subjects Characteristics	AD (Mean ± SD)	HD (Mean ± SD)	*f*-Statistic	Coefficient (*β*)	*p* Value (Changes between the Group Repeated)		
					Intervention	BMI	Sex
BMI, kg/m^2^	31.39 ± 3.27	31.45 ± 3.20	0.01	0.001	0.99	-	0.58
Systolic blood pressure, mmHg	132.55 ± 12.63	132.64 ± 13.10	0.16	−7.55	0.16	0.12	0.40
Diastolic blood pressure, mmHg	77.94 ± 10.38	76.37 ± 8.71	0.63	1.14	0.37	0.13	0.14
Fasting blood glucose, mmol/L	5.29 ± 0.57	5.31 ± 0.59	0.22	−0.11	0.63	0.25	0.99
Fasting blood insulin, pmol/L	124.34 ± 70.81	129.86 ± 73.59	1.52	−13.71	0.93	0.01 *	0.39
Insulin resistance, HOMA-IR	5.20 ± 3.80	5.19 ± 3.35	0.42	−0.28	0.66	0.02 *	0.26
HDL, mmol/L	1.08 ± 0.24	1.09 ± 0.26	0.11	0.03	0.72	0.01 *	0.53
LDL, mmol/L	2.50 ± 0.92	2.59 ± 0.96	0.01	−0.054	0.97	0.97	0.36
TG, mmol/L	1.40 ± 0.63	1.49 ± 0.72	0.34	0.007	0.55	0.49	0.02 *
TC, mmol/L	4.35 ± 1.06	4.47 ± 1.12	0.02	−0.07	0.90	0.80	0.04 *

Values are means ± SDs (n = 27). Differences between groups after six weeks were analyzed using a generalized linear mixed model (GLMM) with intervention (AD; ≤2 servings/day of dairy products, HD; ≥4 servings/day of dairy products), sequence (AD/HD, HD/AD), intervention * sequence (carryover effect) as fixed attributes adjusted for sex and BMI with visits (V1, V2) as the repeated statement. HOMA-IR, homeostatic model assessment of insulin resistance; HDL, high-density lipoproteins; LDL, low-density lipoproteins; TG, triglycerides; TC, total cholesterol; * *p* < 0.05. is significant in the legend of table.

**Table 2 nutrients-13-02088-t002:** Coefficient estimates from GLMM analyses for dietary intake at the end of each phase following adequate dairy product intake (AD) and high dairy product intake (HD) in subjects with hyperinsulinemia.

Dietary Intake	AD (Mean ± SD)	HD (Mean ± SD)	*f*-Statistic	Coefficient (*β*)	*p* Value (Changes between the Group Repeated)		
					Intervention	BMI	Sex
Total energy intake (kcal/d)	2097.88 ± 761.73	2438.66 ± 887.63	6.15	−292.07	0.01 *	0.18	0.006 *
Total carbohydrate intake (g/d)	236.62 ± 95.56	273.62 ± 109.27	12.43	−33.29	0.04 *	0.34	0.01 *
Total protein intake (g/d)	91.51 ± 33.15	116.48 ± 35.64	2.13	−21.65	0.001 *	0.83	0.12 *
Total fat intake (g/d)	84.13 ± 33.15	95.06 ± 38.84	9.58	−8.75	0.01 *	0.09	0.005 *
Saturated fatty acids (g/d)	28.76 ± 12.63	38.73 ± 16.62	0.80	−8.56	0.002 *	0.10	0.007 *
Polyunsaturated fatty acids (g/d)	15.04 ± 5.82	14.15 ± 6.10	0.29	1.72	0.31	0.19	0.10
Monounsaturated fatty acids (g/d)	34.35 ± 14.02	34.91 ± 15.01	1.35	−1.69	0.52	0.10	0.002 *
Arachidonic acid (AA) (g/d)	0.13 ± 0.06	0.12 ± 0.05	3.62	−0.01	0.24	0.18	0.12
Eicosapentaenoic acid (EPA) (g/d)	0.21 ± 0.29	0.19 ± 0.28	4.47	0.02	0.60	0.84	0.48
Docosahexaenoic acid (DHA) (g/d)	0.23 ± 0.26	0.21 ± 0.22	3.62	0.4	0.63	0.76	0.51

Values are means ± SDs (n = 27.). Differences between groups after six weeks were analyzed using a generalized linear mixed model (GLMM) with intervention (AD; ≤2 servings/day of dairy products, HD; ≥4 servings/day of dairy products), sequence (AD/HD, HD/AD), intervention * sequence (carryover effect) as fixed attributes adjusted for sex and BMI with visits (V1, V2) as the repeated statement. Dietary intake was extracted from food frequency questionnaire (FFQ). * *p* < 0.05 is significant in the legend of table.

**Table 3 nutrients-13-02088-t003:** Coefficient estimates from GLMM analyses for F_2_-Isoprostane isomers following adequate dairy product intake (AD) and high dairy product intake (HD) in subjects with hyperinsulinemia.

Isoprostanes (IsoPs) (ng/mL)	*f*-Statistic	Coefficient (*β*)	*p* Value (Changes between the Group Repeated)		
			Intervention	BMI	Sex
5(*RS*)-5-F_2***c***_-IsoP	1.328	0.061	0.255	0.677	0.001 *
5-*epi*-5-F_2***t***_-IsoP	2.520	−0.008	0.119	0.571	0.001 *
5-F_2***t***_-IsoP	11.215	0.023	0.002 *	0.090	<0.001 *
8-F_2***t***_-IsoP	8.961	0.003	0.004 *	0.259	<0.001 *
15-*epi*-15-F_2***t***_-IsoP	2.367	0.005	0.131	0.551	0.002 *
15-F_2***t***_-IsoP	3.736	0.006	0.059	0.014 *	0.900
Total F_2_-IsoPs	4.818	0.089	0.033 *	0.326	0.001 *

n = 27. Differences between groups after six weeks were analyzed using a generalized linear mixed model (GLMM) with intervention (AD, HD), sequence (AD/HD, HD/AD), intervention * sequence (carryover effect) as fixed attributes adjusted for sex and BMI with visits (V1, V2) as the repeated statement. * *p* < 0.05 significant.

## Data Availability

Data sharing is not applicable to this article.

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
