# Peer review of "Impact of Dairy Intake on Plasma F2-IsoProstane Profiles in Overweight Subjects with Hyperinsulinemia: A Randomized Crossover Trial"

_nutrients, 2021, doi:10.3390/nu13062088_

Round 1

Reviewer 1 Report

The authors aimed to investigate the impact of dairy intake on plasma F2-IsoProstane profiles in overweight subjects with hyperinsulinemia.

The study is original and interesting but there are some methodological concerns and the results analysis is not clear. Thus, the text needs major improvements.

The literature review needs clarification to support the hypothesis. In addition, the English makes several sentences difficult to understand or even incorrect (e.g.: through peroxidation of membrane lipids such as LDL-cholesterol).

Regarding the design, one of my concerns is the age range of participants (28-71 years) and the inclusion of participants of both sexes in regard to the eventual sex dependent prostaglandin metabolism. The sex issue was controlled with the statistical analysis but, once confirmed, statistical analysis should not consider participants as a whole. I cannot find any result regarding age, although the authors report having performed this analysis in the statistics section. One other concern regarding the participants’ description is that they are poorly characterized, namely the title states “overweight subjects” and there is no reference to this in the participants section. Although the inclusion criteria states “fasting insulin >90 pmol/L, fasting glucose <7mmol/L, glycated hemoglobin (HbA1c) <6.5%.”, participants are not definitely characterized as hyperinsulinemic non diabetics or the ethiology of the hyperinsunemia described. Exclusion criteria states “(approximately > 2 servings/d)” - what was exactly the cut off value? – and “lactose or gluten tolerance,” – subjects tolerant to lactose were excluded?

Still concerning the study design, the interventions were not controlled for energy intake, which resulted in a higher energy intake in the HD intervention.

Regarding the methods section, instruments for measuring weight and height were not reported. Quantification of F2-IsoPs is usually performed in plasma/serum and not in blood, as reported. Plasma usage is confirmed in figure 2 in the results. Determination of HDL-C is poorly described.

If glycated haemoglobin was used as a criteria for selecting participants, the methodology used to evaluate it should be described and the results presented.

Statistical analysis is not clear. Apparently several statistical tests were used to test the same hypothesis, namely parametric and non-parametric pared tests were used to compare the groups. The GLM with repeated measures was used in addition to t-test and Wilkoxon test.

Results

Taking into account the statistical analysis performed (generalized linear mixed model (GLMM) with repeated measures), if values at the end of the intervention are presented, values at baseline should also be presented, or alternatively the changes – as presented in figure 2.

There are several inconsistencies in the reporting of results (e.g.: The mean of dairy products consumption after HD compared with AD was 2.4 ± 1.2 and 5.8 ± 2.1 servings/d, respectively.” – HD consumed less then AD?; or the authors refer body fat mass but this variable was not previously described).

The nutritional information available for EPA and DHA presence in food items is often scarce and results regarding these nutrients intake should be discussed with this in mind.

Once sex was identified as a confounding variable, groups should have been separated for posterior analysis of intervention effect – this is not clear in the text. Once separated by sex and eliminated the sequence effect, the effect of time (pre-post – with-in subjects effect) and the intervention (HD/AD – between groups effect) can only be considered using the GML model if there is no interaction between these factors. In order to infer a difference in the changes between interventions, it’s necessary to have an interaction effect. Maybe this is why t-test were used but the rationale for the statistical analysis is not clear neither in the methods nor the results sections.

Thus, with these many doubts, I cannot make a conclusive analyse of the results or the discussion.

In addition, in the discussion, the term “decreased” is very confusing. From figure 2, none of the changes observed from pre to post intervention were negative, so there were no decreases. Increases, if significant, were larger or smaller.

Minor Suggestions:

Abstract:

Line 17: “may modified” à consider: may be modified

Line 18: “high dairy product (HD) consumption à consider replacing product with products

Line 18: “adequate dairy (AD)” à consider: adequate dairy products (AD) consumption

Line 20: “in two phases” à consider: in two groups

Line 20: “6 weeks” à 6 weeks.

Line 21: “blood glucose (FBG) and insulin, homeostatic model assessment” à consider: blood glucose (FBG), insulin, and homeostatic model assessment

Line 22: “Isomers of F2-IsoPs were quantified by HPLC-MS/MS.” Which isomers?

Line 26: “After HD compared to AD, a lower level of F2-IsoPs in women compared to men was observed.” The sentence needs clarifying

Introduction:

Line 37: “F2-IsoPs are believed to act through [2], impairment of glycemic homeostasis [3], stimulation of proliferative responses in fibroblasts [4] and alterations of membrane lipids [5].” à it necessary to discriminate what ref 2 indicates as the mechanism of action

Line 39: “Moreover, this association between F2-IsoPs, glycemic homeostasis and dietary intake, especially dairy products have not been well studied.” Consider replacing this with the

Line 41: associated to àassociated with

Line 43: “The F2-IsoPs have been implicated in the impairment of beta-cell functions and to induce apoptosis” à consider: and seem to induce or and have been shown to induce

Line 44: “(…) through peroxidation of membrane lipids such as LDL-cholesterol [6]” à LDL-cholesterol is not a membrane lipid

Line 44-46: “confirmed an increase in total levels of F2-IsoPs (…) among diabetic patients in comparison to non-diabetic [3-6] and obese adults [7].”  Comparison of diabetic with obese?

Line 48: “(…) determine the incidence of T2D among healthy and obese populations [2]”. If the subjects are healthy they are not supposed to be diabetic

Line 49-50: “The dietary intake, specifically of dairy products could contribute to the prevention of oxidative stress and an adequate glycemic response.” What are the evidences supporting this sentence?

Author Response

Comments and Suggestions for Authors

The authors aimed to investigate the impact of dairy intake on plasma F2-IsoProstane profiles in overweight subjects with hyperinsulinemia.

The study is original and interesting, but there are some methodological concerns and the analysis of the results is not clear. Thus, the text needs major improvements.

  1. The literature review needs clarification to support the hypothesis.

Response: Thanks for your comments, more studies were added to support the hypothesis of the study (at lines 55-63). Furthermore, we reworded the hypothesis at the lines 57-62 as follows: “Despite indirect evidence of the potential role of dairy products in oxidative stress and glycemic parameters, the association between T2D risk factors and F2-IsoPs after dairy product intake remains unknown in overweight individuals with hyperinsulinemia. Thus, the hypothesis of this study is that the level of F2-IsoPs will be modified after consumption of high dairy (HD) products compared with adequate dairy (AD) products.”

  1. In addition, the English makes several sentences difficult to understand or even incorrect (e.g.: through peroxidation of membrane lipids such as LDL-cholesterol).

Response: The sentence was edited (lines 43-44), the manuscript was edited for grammatical errors and the manuscript has been checked thoroughly.

  1. Regarding the design, one of my concerns is the age range of participants (28-71 years) and the inclusion of participants of both sexes in regard to the eventual sex-dependent prostaglandin metabolism. The sex issue was controlled with the statistical analysis, but, once confirmed, statistical analysis should not consider participants as a whole. I cannot find any result regarding age, although the authors report having performed this analysis in the statistics section.

Response: Thanks for your question about age and sex, which are a very important factor to be considered,

  • We did four (4) models of GLMM with the subject as the random factor, including:
  • First, Fixed effects: Intervention, visits, sequence, Intervention * sequence, age (p>0.05), sex, and BMI,
  • Second, Fixed effects: Intervention, visits, sequence, Intervention * sequence, age (p>0.05) and BMI,
  • Third, Fixed effects: Intervention, visits, sequence, Intervention * sequence, age (p>0.05), and sex
  • Fourth, Fixed effects: Intervention, visits, sequence, Intervention * sequence sex and BMI

For these 4 models, no significant effect of age was observed on total F2-Isoprostane or on specific isomers. Therefore, we removed the fixed factor age from our analysis to obtain a better model fit. We used the fourth model to report on our results. But we added a sentence in the results to explain that there was no effect of age on F2-IsoPs (lines 222-225).

  1. One other concern regarding the participants’ description is that they are poorly characterized, namely the title states “overweight subjects” and there is no reference to this in the participant’s section.

Response: The BMI of all participants was higher than 25 kg/m2 which was considered as overweight or obese (Base on data, there are, 7 overweight 25 ≤ BMI≤ 29.9kg/m2) and 20 obese BMI ≥ 30 kg/m2) subjects in our study). We added more detailed in the description results of participants for BMI (lines 174-175 & lines 72-73)

  1. Although the inclusion criteria states “fasting insulin >90 pmol/L, fasting glucose <7mmol/L, glycated hemoglobin (HbA1c) <6.5%.”, participants are not definitely characterized as hyperinsulinemic non-diabetics or the etiology of the hyperinsulinemia described.

Response: The main important inclusion criteria for the participants’ study were hyperinsulinemia, in which all participants were selected based on fasting insulin >90 pmol/L, fasting glucose <7.0 mmol/L, glycated hemoglobin (HbA1c) <6.5% (lines 73-74).

Subject selection: Inclusion criteria: Caucasian men and postmenopausal women (absence of menstrual cycles for >12 months) aged >18 yrs; BMI between 25-40 kg/m2; hyperinsulinemia (fasting plasma insulin >90 pmol/l, which represents >75th percentile for fasting insulin levels in a sample from the adult Quebec population Not always the case in Table 1 because of high SD); fasting plasma glucose (FPG) <7.0 mmol/l; HbA1c <6.5% (47;48)); if treated with lipid-lowering agents, the dose must have been stable over the last 3 months; stable body weight (±5%) for 3 months; willing to consume study foods and able to follow protocol and give informed consent.

  1. Exclusion criteria states “(approximately > 2 servings/d)” - what was exactly the cut off value? – and “lactose or gluten tolerance,” – subjects tolerant to lactose were excluded?

Response: The cut off for high-dairy intervention in our study was an intake of 4 or more than 4 servings/day, and for adequate-dairy intervention was an intake of 2 or less than 2 servings/day, however, if participants consumed high dairy intake (³2 servings/day), she/he could not participate to the study (exclusion criteria), we edited the sentence to make it clearer (Lines 75-76).

In addition, if participants had an allergy, intolerance, or dislike to dairy consumption or nutrient absorption problems (such as lactose intolerance)- they were excluded from the study. Please see below specific exclusion criteria.

Exclusion criteria: Failure to meet any one or more of the inclusion criteria; a diagnosis of T2D; high dairy consumption (³2 servings/day), major surgery in the 3 months prior to study onset, smoking, incompatibility with dairy consumption (allergy, intolerance or dislike), inflammatory bowel disease or other gastrointestinal disorder influencing gastrointestinal motility or nutrient absorption; medications known to affect lipid and glucose metabolism other than those used to treat hypertension or dyslipidemia; diseases known to affect glucose metabolism. (Lines 74-81)

  1. Still concerning the study design, the interventions were not controlled for energy intake, which resulted in a higher energy intake in the HD intervention.

Response: We checked the collinearity test based on linear regression and we found there was non-significant collinearity (p = 0.36) between BMI and energy intake; therefore, we reanalyze the generalized linear regression model (GLMM) and consider energy intake, sex, and BMI as fixed effect parameters, based on the table below there was no effect on energy intake on isoprostane levels. As we mentioned before, similar results were found for age (we added this result to the manuscript, in lines 223-226).

Therefore, we removed energy intake and age from the model, and just sex and BMI were considered as fixed effect parameters in GLMM.

Isoprostanes (IsoPs)

(ng/ml)

f-statistic

Coefficient

(β)

p-value

(Sequence)

p-value

(Changes between the group repeated)

Intervention

BMI

Sex

age

Energy intake

5(RS)-5-F2c-IsoP

1.147

0.061

0.504

0.290

0.667

0.066

0.925

0.633

5-epi-5-F2t-IsoP

1.118

-0.008

0.522

0.296

0.979

0.004*

0.515

0.103

5-F2t-IsoP

11.562

0.023

0.843

0.001*

0.159

0.012*

0.669

0.531

8-F2t-IsoP

7.413

0.003

0.422

0.009*

0.267

0.015*

0.712

0.895

15-epi-15-F2t-IsoP

1.352

0.005

0.59

0.251

0.954

0.007*

0.194

0.363

15-F2t-IsoP

2.107

0.006

0.157

0.154

0.012*

0.403

0.456

0.117

Total F2-IsoPs

3.379

0.089

0.627

0.073

0.512

0.011*

0.665

0.775

*Significant p< 0.05

  1. Regarding the methods section, instruments for measuring weight and height were not reported.

Response: Thank you, we have added these details at lines 105-110: “Body weight was measured with a professional scale accurate to 0.1 kg (Health O Meter Professional, Sunbeam products, Inc.) and height was measured using a wall-mounted stadiometer with 1 mm accuracy (The Easy-Glide Bearing Stadiometer, Perspective Enterprises), with subjects in light clothing and without shoes. Body mass index (BMI) was calculated as weight (kg) divided by height (m) squared and recorded as kg/m2. Body composition was evaluated, in the fasting state and at the same time across visits, using a 4-electrode bioimpedance scale (InBody 520 Body Composition Analyzer).

  1. Quantification of F2-IsoPs is usually performed in plasma/serum and not in blood, as reported.

Response: Thanks for your comment, in the method section we reported the plasma level of F2-IsoPs, and in the whole manuscript the level of F2-Isops is now always reported in plasma. (Lines 126, 128, 140)

  1. Plasma usage is confirmed in figure 2 in the results.

Response: Thanks for noticing. We added plasma levels of F2-IsoPs in the legend of figure 2. (Lines 220)

  1. Determination of HDL-C is poorly described.

Response: We added more details at lines 121-123) as follows: “The HDL cholesterol (HDL-C) fraction was obtained after precipitation of very low-density lipoprotein and LDL particles in the infranatant with heparin manganese chloride [18]. LDL cholesterol (LDL-C) was calculated with the Friedewald formula [19]. LDL-C was calculated with the Fried Ewald formula [20]. “(Line 121-125)

  1. If glycated haemoglobin was used as a criterion for selecting participants, the methodology used to evaluate it should be described and the results presented.

Response: We measured HbA1c at the beginning of the study (screening) and we added the screening measurement of HbA1c at line 172).

  1. Statistical analysis is not clear. Apparently, several statistical tests were used to test the same hypothesis,
  • namely parametric and non-parametric pared tests were used to compare the groups.

Response: Yes, it is true because often reviewers like to see both, parametric and non-parametric tests when data are normally or non-normally distributed within the same table. This was done only for clinical parameters of tables 1 and 2. The conclusion remained the same for both tests though (Tables 1 and 2). We have modified at line 156 to add “both,”,

  • The GLM with repeated measures was used in addition to t-test and Wilcoxon test.

Response: Thanks for your comment. In the current paper, we used only the generalized linear mixed model (GLMM) (not a general linear model (GLM)). GLMM analysis is an extension and a more powerful procedure than the GLM. Indeed, GLMM uses both, fixed and random effects. The subject was considered a random effect in this study. Multiple comparisons within the GLMM were corrected with Bonferroni.

Results

  1. Taking into account the statistical analysis performed (generalized linear mixed model (GLMM) with repeated measures), if values at the end of the intervention are presented, values at baseline should also be presented, or alternatively the changes – as presented in figure 2.

Response: Some authors use the difference with the baseline and compare this difference with a t-test. We chose a more powerful approach with the GLMM. This baseline would not be informative since we don’t have a sequence effect or sequence*intervention (AB vs BA) showing a carry-over in the model. We add more explanations, “3.3 the Isomers of F2-IsoPs” (lines 225-227).

Furthermore, we add another supplementary table, TableS1 (modified supplementary table file was attached), which showed using Wilcoxon and paired t-test, there were no difference in the level of total F2-IsoPs (p < 0.05) and isomers of F2-IsoPs at the beginning of each phase including adequate-dairy and high-dairy

  1. There are several inconsistencies in the reporting of results (e.g.: The mean of dairy products consumption after HD compared with AD was 2.4 ± 1.2 and 5.8 ± 2.1 servings/d, respectively.”

 HD consumed less then AD?

Response: Thanks for the comment, we reversed the order (lines188-189)

  1. the authors refer body fat mass but this variable was not previously described).

Response: Thanks, we added measurement of body composition to the “anthropometric assessment” section of the method (lines 112-113)

  1. The nutritional information available for EPA and DHA presence in food items is often scarce and results regarding these nutrients intake should be discussed with this in mind.

Thanks for your comments, we agree there is a gap of knowledge. We edited (line 283) to put the level EPA/DHA in the context of the plasma level. Furthermore, FFQ has limitations to report the intake of micronutrients and macronutrients precisely, here we summarized the advantage and disadvantage of FFQ:

Advantage:

  • Lower administrative costs and time
  • Ability to assess usual and long-term intake

           Disadvantage:

  • Inaccuracy of absolute nutrient values,
  • Fluctuation of nutrient values depending on instrument length and structure,
  • Lack of detail regarding specific foods

We added the advantages and disadvantages of using FFQ in the section of limitation and strength. (Lines 312-313)

  • Goulet J, Nadeau G, Lapointe A, Lamarche B, Lemieux S. Validity and reproducibility of an interviewer-administered food frequency questionnaire for healthy French-Canadian men and women. Nutr J 2004,3,13.

  1. Once sex was identified as a confounding variable, groups should have been separated for posterior analysis of intervention effect – this is not clear in the text. Once separated by sex and eliminated the sequence effect, the effect of time (pre-post – with-in subjects’ effect) and the intervention (HD/AD – between groups effect) can only be considered using the GML model if there is no interaction between these factors. In order to infer a difference in the changes between interventions, it’s necessary to have an interaction effect. Maybe this is why t-test were used but the rationale for the statistical analysis is not clear neither in the methods nor the results sections. Thus, with these many doubts, I cannot make a conclusive analyse of the results or the discussion.

Response: To reassure the reviewer: It is exactly what we did inside the global GLMM model and not separately as two analyses one for men and one woman (the sample size was too low and would have affected the power of the analysis). Under these circumstances, we preferred a single global model that takes everything into account. In figure 2, we used the global GLMM model. Since we have a significant effect of sex (there are no covariates only factors in GLMM), we plotted the marginal estimated means for men and women (predicted value) for each F2-isoPs provided by the overall model (not a separated analysis and the other effects were taken into account in estimated means). 

  1. In addition, in the discussion, the term “decreased” is very confusing.

We edited and replaced with the proper words, “lower” and “was reduced” respectively (lines 247 and 305)

  1. From figure 2, none of the changes observed from pre to post intervention were negative, so there were no decreases. Increases, if significant, were larger or smaller.

Response: Figure 2 reported changes between adequate (gray bars) vs high-dairy diet (white bar according to sex). The crossover design here looks at the treatment and pre-/post is taken into account, no matter which one was started first. The significance of contrast in the figure does not indicate which we have now corrected.  

Minor Suggestions:

  1. Abstract:

Line 17: “may modified” à consider: may be modified

The correction was done, thank you (line 17)

Line 18: “high dairy product (HD) consumption à consider replacing product with products

The correction was done (line 18)

Line 18: “adequate dairy (AD)” à consider: adequate dairy products (AD) consumption

The correction was done (line 19)

Line 20: “in two phases” à consider: in two groups

The correction was done (line 20)

Line 20: “6 weeks” à 6 weeks.

The correction was done (line 21)

Line 21: “blood glucose (FBG) and insulin, homeostatic model assessment” à consider: blood glucose (FBG), insulin, and homeostatic model assessment

The correction was done (line 21)

Line 22: “Isomers of F2-IsoPs were quantified by HPLC-MS/MS.” Which isomers?

The correction was done (line 22)

Line 26: “After HD compared to AD, a lower level of F2-IsoPs in women compared to men was observed.” The sentence needs clarifying

 All of the corrections were done (line 27)

  1. Introduction:

Line 37: “F2-IsoPs are believed to act through [2], impairment of glycemic homeostasis [3], stimulation of proliferative responses in fibroblasts [4] and alterations of membrane lipids [5].” à it necessary to discriminate what ref 2 indicates as the mechanism of action

The correction was done (We deleted the unrelated reference)

Line 39: “Moreover, this association between F2-IsoPs, glycemic homeostasis and dietary intake, especially dairy products have not been well studied.” Consider replacing this with the

The correction was done (#39)

Line 41: associated to associate with

The correction was done (#41)

Line 43: “The F2-IsoPs have been implicated in the impairment of beta-cell functions and to induce apoptosis” à consider: and seem to induce or and have been shown to induce

The correction was done (#42-43)

Line 44: “(…) through peroxidation of membrane lipids such as LDL-cholesterol [6]” à LDL-cholesterol is Not a membrane lipid

The correction was done (#44)

Line 44-46: “confirmed an increase in total levels of F2-IsoPs (…) among diabetic patients in comparison to non-diabetic [3-6] and obese adults [7].”  Comparison of diabetic with obese?

… comparison to healthy overweight adults (#46-47)

Line 48: “(…) determine the incidence of T2D among healthy and obese populations [2]”. If the subjects are healthy, they are not supposed to be diabetic

The correction was done

Line 49-50: “The dietary intake, specifically of dairy products could contribute to the prevention of oxidative stress and an adequate glycemic response.” What is the evidence supporting this sentence?

We add mechanisms (#50-51)

Reviewer 2 Report

Impact of dairy intake on plasma F2-IsoProstane profiles in overweight subjects with hyperinsulinemia  is a very nice article. Some considerations

The main circumstance is: I know that you have used adequate dairy  in another article but it is not correct to consider less than 2 servings of dairy per day adequate dairy, it is preferable to use another term.

In fact you in the discussion go along that line by saying “Generally, the level of F2-IsoPs decreased after consumption of HD compared to AD, 233 especially total level of F2-IsoP, 5-F2t-IsoP, 8-F2t-IsoP, and 15-F2t-IsoP. Similarly, a random-234 ized-parallel-group study among obese adults with metabolic syndrome indicated a re-235 duction in a plasma after intake of adequate-dairy (3.5 servings/d) compared to low-dairy 236 (<0.5 daily servings) during 12-week of intervention [22]”

Other minor considerations:

  1. in the abstract after for 6-week , yYou have to put a point
  2. In Methods :All subjects had hyperinsulinemia and 70 were selected based on fasting insulin >90 pmol/L, fasting glucose <7.0 mmol/L, glycated 71 hemoglobin (HbA1c) <6.5%.

In the inclusion does not refer to being overweight, which must be included.

  1. On page 238, please delete an in

Author Response

Comments and Suggestions for Authors

Impact of dairy intake on plasma F2-IsoProstane profiles in overweight subjects with hyperinsulinemia is a very nice article. Some considerations

The main circumstance is: I know that you have used adequate dairy in another article but it is not correct to consider less than 2 servings of dairy per day adequate dairy, it is preferable to use another term. In fact, you in the discussion go along that line by saying “Generally, the level of F2-IsoPs decreased after consumption of HD compared to AD, especially total level of F2-IsoP, 5-F2t-IsoP, 8-F2t-IsoP, and 15-F2t-IsoP. Similarly, a randomized-parallel-group study among obese adults with metabolic syndrome indicated a reduction in a plasma after intake of adequate-dairy (3.5 servings/d) compared to low-dairy (<0.5 daily servings) during 12-week of intervention [22]”.

Thanks for your comments, the study was designed (the study was started in 2017) based on previous Canadian dietary guideline (2017) where the recommended daily dairy servings for healthy adults was 2-4 servings/day; therefore, based on the guideline recommendation, we considered 2 or less than 2 serving/day as adequate dairy product and 4 and higher than 4 serving/day as the high-dairy product (line 75).    

Other minor considerations:

  1. in the abstract after for 6-week, you have to put a point

The correction was done (Line 21)

  1. In Methods: All subjects had hyperinsulinemia and were selected based on fasting insulin >90 pmol/L, fasting glucose <7.0 mmol/L, glycated hemoglobin (HbA1c) <6.5%. In inclusion does not refer to being overweight, which must be included.

We added more explanation (Lines 71-73)

BMI between 25-40 kg/m2 (overweight, 25 ≤ BMI≤ 29.9 kg/m2 or obesity, BMI ≥ 30 kg/m2) had hyperinsulinemia

  1. On page 238, please delete an in

The correction was done

We move sentences, therefore, the new line number is 53

Round 2

Reviewer 1 Report

The authors answer revels that most of my concerns regarding the study design and the statistical analysis have been addressed. However, these answer did not translate into actual changes of the text in order to make it clear to the reader. Most of the information that was given is still missing, namely the sex and energy intake control analysis.

Namely:

  1. Regarding the design, one of my concerns is the age range of participants (28-71 years) and the inclusion of participants of both sexes in regard to the eventual sex-dependent prostaglandin metabolism. The sex issue was controlled with the statistical analysis, but, once confirmed, statistical analysis should not consider participants as a whole. I cannot find any result regarding age, although the authors report having performed this analysis in the statistics section.

Response: Thanks for your question about age and sex, which are a very important factor to be considered,

  • We did four (4) models of GLMM with the subject as the random factor, including:
  • First, Fixed effects: Intervention, visits, sequenceIntervention * sequenceage (p>0.05)sex, and BMI,
  • Second, Fixed effects: Intervention, visits, sequenceIntervention * sequence, age (p>0.05) and BMI,
  • Third, Fixed effects: Intervention, visits, sequenceIntervention * sequence, age (p>0.05), and sex
  • Fourth, Fixed effects: Intervention, visits, sequenceIntervention * sequence sex and BMI

For these 4 models, no significant effect of age was observed on total F2-Isoprostane or on specific isomers. Therefore, we removed the fixed factor age from our analysis to obtain a better model fit. We used the fourth model to report on our results. But we added a sentence in the results to explain that there was no effect of age on F2-IsoPs (lines 222-225).

Regarding your answer, you refer you used model 4 and yet in the text you state:

“The fixed effects were intervention (HD, AD), visit number (V2, 161 V4), sequence (AD/HD, HD/AD), age, and BMI.”

  1. Still concerning the study design, the interventions were not controlled for energy intake, which resulted in a higher energy intake in the HD intervention.

Response: We checked the collinearity test based on linear regression and we found there was non-significant collinearity (p = 0.36) between BMI and energy intake; therefore, we reanalyze the generalized linear regression model (GLMM) and consider energy intake, sex, and BMI as fixed effect parameters, based on the table below there was no effect on energy intake on isoprostane levels. As we mentioned before, similar results were found for age (we added this result to the manuscript, in lines 223-226).

In the statistical analysis there is no reference to:

We checked the collinearity test based on linear regression and we found there was non-significant collinearity (p = 0.36) between BMI and energy intake

  1. If glycated haemoglobin was used as a criterion for selecting participants, the methodology used to evaluate it should be described and the results presented.

Response: We measured HbA1c at the beginning of the study (screening) and we added the screening measurement of HbA1c at line 172).

I was referring to introducing information in the methods section, although this information is pertinent too.

  1. Statistical analysis is not clear. Apparently, several statistical tests were used to test the same hypothesis,
  • namely parametric and non-parametric pared tests were used to compare the groups.

Response: Yes, it is true because often reviewers like to see both, parametric and non-parametric tests when data are normally or non-normally distributed within the same table. This was done only for clinical parameters of tables 1 and 2. The conclusion remained the same for both tests though (Tables 1 and 2). We have modified at line 156 to add “both,”,

This is an option (that wouldn´t be my, but I respect). However, if the Authors go this way they should inform the reader which variables were normally distributed and which were not.

  • The GLM with repeated measures was used in addition to t-test and Wilcoxon test.

Response: Thanks for your comment. In the current paper, we used only the generalized linear mixed model (GLMM) (not a general linear model (GLM)). GLMM analysis is an extension and a more powerful procedure than the GLM. Indeed, GLMM uses both, fixed and random effects. The subject was considered a random effect in this study. Multiple comparisons within the GLMM were corrected with Bonferroni.

That is exactly my point. Why use t-test and Wilcoxon if the GLMM can give you the results of the comparisons. Unless there is an interaction between the factors that was not reported.

  1. Taking into account the statistical analysis performed (generalized linear mixed model (GLMM) with repeated measures), if values at the end of the intervention are presented, values at baseline should also be presented, or alternatively the changes – as presented in figure 2.

Response: Some authors use the difference with the baseline and compare this difference with a t-test. We chose a more powerful approach with the GLMM. This baseline would not be informative since we don’t have a sequence effect or sequence*intervention (AB vs BA) showing a carry-over in the model. We add more explanations, “3.3 the Isomers of F2-IsoPs” (lines 225-227).

My interpretation is that: as you do not have a sequence effect or sequence * intervention effect you can retrieve information from these results. If you had, you could not interpret pre/post values and you would have to explore the changes.

Furthermore, we add another supplementary table, TableS1 (modified supplementary table file was attached), which showed using Wilcoxon and paired t-test, there were no difference in the level of total F2-IsoPs (p < 0.05) and isomers of F2-IsoPs at the beginning of each phase including adequate-dairy and high-dairy

  1. The nutritional information available for EPA and DHA presence in food items is often scarce and results regarding these nutrients intake should be discussed with this in mind.

Thanks for your comments, we agree there is a gap of knowledge. We edited (line 283) to put the level EPA/DHA in the context of the plasma level. Furthermore, FFQ has limitations to report the intake of micronutrients and macronutrients precisely, here we summarized the advantage and disadvantage of FFQ:

Advantage:

  • Lower administrative costs and time
  • Ability to assess usual and long-term intake

Disadvantage:

  • Inaccuracy of absolute nutrient values,
  • Fluctuation of nutrient values depending on instrument length and structure,
  • Lack of detail regarding specific foods

We added the advantages and disadvantages of using FFQ in the section of limitation and strength. (Lines 312-313)

  • Goulet J, Nadeau G, Lapointe A, Lamarche B, Lemieux S. Validity and reproducibility of an interviewer-administered food frequency questionnaire for healthy French-Canadian men and women. Nutr J 2004,3,13.

The sentence the Authors introduced, although pertinent, is quite general. One should reinforce that in spite of the ability of FFQs to evaluate long-term consumption of energy and some nutrients, regarding some micronutrients as essential fatty acids, the information available for the composition of a lot of food items is scarce. So in this particular case, the probability of underestimation is quite high.

  1. Once sex was identified as a confounding variable, groups should have been separated for posterior analysis of intervention effect – this is not clear in the text. Once separated by sex and eliminated the sequence effect, the effect of time (pre-post – with-in subjects’ effect) and the intervention (HD/AD – between groups effect) can only be considered using the GML model if there is no interaction between these factors. In order to infer a difference in the changes between interventions, it’s necessary to have an interaction effect. Maybe this is why t-test were used but the rationale for the statistical analysis is not clear neither in the methods nor the results sections. Thus, with these many doubts, I cannot make a conclusive analyse of the results or the discussion.

Response: To reassure the reviewer: It is exactly what we did inside the global GLMM model and not separately as two analyses one for men and one woman (the sample size was too low and would have affected the power of the analysis). Under these circumstances, we preferred a single global model that takes everything into account. In figure 2, we used the global GLMM model. Since we have a significant effect of sex (there are no covariates only factors in GLMM), we plotted the marginal estimated means for men and women (predicted value) for each F2-isoPs provided by the overall model (not a separated analysis and the other effects were taken into account in estimated means).

I do agree with this option but why then keep comparing groups with t-test and Wilcoxon test?

  1. In addition, in the discussion, the term “decreased” is very confusing.

We edited and replaced with the proper words, “lower” and “was reduced” respectively (lines 247 and 305)

The terms “decrease”, “lowered”, “reduction” are also used in the results section to compare groups, namely in lines 202 and …. These terms imply changes in time and not differences between groups in the changes in time.

  1. From figure 2, none of the changes observed from pre to post intervention were negative, so there were no decreases. Increases, if significant, were larger or smaller.

Response: Figure 2 reported changes between adequate (gray bars) vs high-dairy diet (white bar according to sex). The crossover design here looks at the treatment and pre-/post is taken into account, no matter which one was started first. The significance of contrast in the figure does not indicate which we have now corrected.

Sorry, maybe the author did not understand my question because I do not understand the answer.  What I am saying is that I cannot interpret Figure 2. The legend states it refers to changes in plasma levels, but the y axis says “plasma levels”. This should be clarified. If these are changes, changes are positive, thus the values of F2-Isoprotanes increased and did not decrease. There is a difference between increasing significantly less and decreasing. This would compromise the interpretation in the discussion. If these are absolute values, it is important to state at what moment - post intervention (no matter which started first)? – and this would come back to the base line values issue.

In addition to the points already mentioned the statistic session still needs clarification regarding the following sentences:

Lines 152-155: “Comparison of baseline parameters (anthropometric parameters, glycemic and lipid profile, F2-IsoPs levels) at the beginning of each phase of the intervention was conducted using paired t-test. Comparison of anthropometric characteristics and clinical parameters between AD and HD was performed using both, paired t-tests and Wilcoxon signed-rank tests.”

– What where the authors comparing in the first sentence and in the second sentence?

Minor suggestions

Lines 52-55: “For instance, in a randomized-parallel-group study indicated a reduction a plasma level of 15-F2t-IsoP after intake of adequate-dairy (3.5 servings/d and ≥ 1000 mg calcium/d) compared to low-dairy (<0.5 daily servings and ≤600 mg calcium/d) 54 during 12-week of intervention in obese adults with metabolic syndrome [8]. However, a 55 6-month randomized, parallel-group intervention study demonstrated no change in in the 56 plasma level of 15-F2t-IsoP after consumption of 3-5 milk servings/d compared to the con- 57 trol group (0.5-1 serving/d) among overweight men and women [9]. Despite indirect evi- 58 dence of the potential role of dairy products on oxidative stress and glycemic parameters, 59 the association between T2D risk factors and F2-IsoPs after dairy product intake remains 60 unknown in overweight individuals with hyperinsulinemia. ><0.5 daily servings and ≤600 mg calcium/d) during 12-week of intervention in obese adults with metabolic syndrome [8]. However, a 6-month randomized, parallel-group intervention study demonstrated no change in in the plasma level of 15-F2t-IsoP after consumption of 3-5 milk servings/d compared to the control group (0.5-1 serving/d) among overweight men and women [9]. Despite indirect evidence of the potential role of dairy products on oxidative stress and glycemic parameters, the association between T2D risk factors and F2-IsoPs after dairy product intake remains unknown in overweight individuals with hyperinsulinemia.”

Consider replacing with: For instance, a randomized-parallel-group study indicated a reduction in plasma level of 15-F2t-IsoP after intake of adequate-dairy (3.5 servings/d and ≥ 1000 mg calcium/d) compared to low-dairy (<0.5 daily servings and ≤600 mg calcium/d) 54 during 12-week of intervention in obese adults with metabolic syndrome [8]. However, a 55 6-month randomized, parallel-group intervention study demonstrated no change in in the 56 plasma level of 15-F2t-IsoP after consumption of 3-5 milk servings/d compared to the con- 57 trol group (0.5-1 serving/d) among overweight men and women [9]. Despite indirect evi- 58 dence of the potential role of dairy products on oxidative stress and glycemic parameters, 59 the association between T2D risk factors and F2-IsoPs after dairy product intake remains 60 unknown in overweight individuals with hyperinsulinemia. ><0.5 daily servings and ≤600 mg calcium/d) during 12-week of intervention in obese adults with metabolic syndrome [8]. However, a 6-month randomized, parallel-group intervention study demonstrated no change in the plasma level of 15-F2t-IsoP after consumption of 3-5 milk servings/d compared to the control group (0.5-1 serving/d) among overweight men and women [9]. Despite indirect evidence of the potential role of dairy products on oxidative stress and glycemic parameters, the association between T2D risk factors and F2-IsoPs after dairy product intake remains unknown in overweight individuals with hyperinsulinemia.

Line 166: “(…) a p-value of less than 0.1 was considered a tendency.” Consider replacing with: a p-value of between 0.05 and 0.1 was considered a tendency

Line 178: “(…)) after AD and HD consumption (…)” consider replacing with: after AD or HD consumption

Author Response

Comments and Suggestions for Authors

The authors answer revels that most of my concerns regarding the study design and the statistical analysis have been addressed. However, these answers did not translate into actual changes of the text in order to make it clear to the reader. Most of the information that was given is still missing, namely the sex and energy intake control analysis.

Namely:

  1. Regarding the design, one of my concerns is the age range of participants (28-71 years) and the inclusion of participants of both sexes in regard to the eventual sex-dependent prostaglandin metabolism. The sex issue was controlled with the statistical analysis, but, once confirmed, statistical analysis should not consider participants as a whole. I cannot find any result regarding age, although the authors report having performed this analysis in the statistics section.

Response: Thanks for your question about age and sex, which are a very important factor to be considered,

  • We did four (4) models of GLMM with the subject as the random factor, including:
  • First, Fixed effects: Intervention, visits, sequenceIntervention * sequenceage (p>0.05)sex, and BMI,
  • Second, Fixed effects: Intervention, visits, sequenceIntervention * sequence, age (p>0.05) and BMI,
  • Third, Fixed effects: Intervention, visits, sequenceIntervention * sequence, age (p>0.05), and sex
  • Fourth, Fixed effects: Intervention, visits, sequenceIntervention * sequence sex and BMI

For these 4 models, no significant effect of age was observed on total F2-Isoprostane or on specific isomers. Therefore, we removed the fixed factor age from our analysis to obtain a better model fit. We used the fourth model to report on our results. But we added a sentence in the results to explain that there was no effect of age on F2-IsoPs (lines 222-225).

Reviewer Comment (Revision 2): Regarding your answer, you refer you used model 4 and yet in the text you state:

“The fixed effects were intervention (HD, AD), visit number (V2, 161 V4), sequence (AD/HD, HD/AD), age, and BMI.”

Response (Revision 2): Thanks for your comment, we replaced age with sex in statistical section Therefore, the fixed effects in GLMM analysis were intervention (HD, AD), visit number (V2, V4), sequence (AD/HD, HD/AD), sex, and BMI.” (Line number 187).”

  1. Still concerning the study design, the interventions were not controlled for energy intake, which resulted in a higher energy intake in the HD intervention.

Response: We checked the collinearity test based on linear regression and we found there was non-significant collinearity (p = 0.36) between BMI and energy intake; therefore, we reanalyze the generalized linear regression model (GLMM) and consider energy intake, sex, and BMI as fixed effect parameters, based on the table below there was no effect on energy intake on isoprostane levels. As we mentioned before, similar results were found for age (we added this result to the manuscript, in lines 223-226).

Reviewer Comment (Revision 2): In the statistical analysis there is no reference to: We checked the collinearity test based on linear regression and we found there was non-significant collinearity (p = 0.36) between BMI and energy intake

Response (Revision 2): Thanks for your comment, we added additional information to the statistical section” The collinearity test was conducted and there was no collinearity between BMI and energy intake (p = 0.36). Therefore, the fixed effects in GLMM analysis were intervention (HD, AD), visit number (V2, V4), sequence (AD/HD, HD/AD), sex, and BMI.” (Line number 185-187)

  1. If glycated haemoglobin was used as a criterion for selecting participants, the methodology used to evaluate it should be described and the results presented.

Response: We measured HbA1c at the beginning of the study (screening) and we added the screening measurement of HbA1c at line 172).

Reviewer Comment (Revision 2): I was referring to introducing information in the methods section, although this information is pertinent too.

Response (Revision 2): Thanks for your comment, we added the method of measurement for HbA1c in the clinical measurements “HbA1c was determined using a colorimetric method after an initial separation by ion exchange chromatography [21]” (Line number 151-152)

  1. Statistical analysis is not clear. Apparently, several statistical tests were used to test the same hypothesis,
  • namely parametric and non-parametric pared tests were used to compare the groups.

Response: Yes, it is true because often reviewers like to see both, parametric and non-parametric tests when data are normally or non-normally distributed within the same table. This was done only for clinical parameters of tables 1 and 2. The conclusion remained the same for both tests though (Tables 1 and 2). We have modified at line 156 to add “both,”,

Reviewer Comment (Revision 2): This is an option (that wouldn´t be my, but I respect). However, if the Authors go this way they should inform the reader which variables were normally distributed and which were not.

Response (Revision 2): We replaced table 1 and 2 with the new analysis of GLMM

  1. The GLM with repeated measures was used in addition to t-test and Wilcoxon test.

Response: Thanks for your comment. In the current paper, we used only the generalized linear mixed model (GLMM) (not a general linear model (GLM)). GLMM analysis is an extension and a more powerful procedure than the GLM. Indeed, GLMM uses both, fixed and random effects. The subject was considered a random effect in this study. Multiple comparisons within the GLMM were corrected with Bonferroni.

Reviewer Comment (Revision 2): That is exactly my point. Why use t-test and Wilcoxon if the GLMM can give you the results of the comparisons. Unless there is an interaction between the factors that was not reported.

Response (Revision 2): Thanks for your comment, based on reviewer comment, we decided to remove the t test and Wilcoxon results and only GLMM used in the paper, moreover we updated the table 1 and 2. And results update (Line number 203-205, 228-235, 301-303

  1. Taking into account the statistical analysis performed (generalized linear mixed model (GLMM) with repeated measures), if values at the end of the intervention are presented, values at baseline should also be presented, or alternatively the changes – as presented in figure 2.

Response: Some authors use the difference with the baseline and compare this difference with a t-test. We chose a more powerful approach with the GLMM. This baseline would not be informative since we don’t have a sequence effect or sequence*intervention (AB vs BA) showing a carry-over in the model. We added more explanations, “3.3 the Isomers of F2-IsoPs” (lines 225-227).

Furthermore, we added another supplementary table, TableS1 (modified supplementary table file was attached), which showed using Wilcoxon and paired t-test, there were no difference in the level of total F2-IsoPs (p < 0.05) and isomers of F2-IsoPs at the beginning of each phase including adequate-dairy and high-dairy

Reviewer Comment (Revision 2): My interpretation is that: as you do not have a sequence effect or sequence * intervention effect you can retrieve information from these results. If you had, you could not interpret pre/post values and you would have to explore the changes.

Response to reviewer 2: Thanks for your comments, in this study there was no sequence effect, or sequence*intervention effect, there based on reviewer comment, we decided to delete sequence columns in table 1, 2, and 3

  1. The nutritional information available for EPA and DHA presence in food items is often scarce and results regarding these nutrients intake should be discussed with this in mind.

Thanks for your comments, we agree there is a gap of knowledge. We edited (line 283) to put the level EPA/DHA in the context of the plasma level. Furthermore, FFQ has limitations to report the intake of micronutrients and macronutrients precisely, here we summarized the advantage and disadvantage of FFQ:

Advantage:

  • Lower administrative costs and time
  • Ability to assess usual and long-term intake

Disadvantage:

  • Inaccuracy of absolute nutrient values,
  • Fluctuation of nutrient values depending on instrument length and structure,
  • Lack of detail regarding specific foods

We added the advantages and disadvantages of using FFQ in the section of limitation and strength. (Lines 312-313)

  • Goulet J, Nadeau G, Lapointe A, Lamarche B, Lemieux S. Validity and reproducibility of an interviewer-administered food frequency questionnaire for healthy French-Canadian men and women. Nutr J 2004,3,13.

Reviewer Comment (Revision 2): The sentence the Authors introduced, although pertinent, is quite general. One should reinforce that in spite of the ability of FFQs to evaluate long-term consumption of energy and some nutrients, regarding some micronutrients as essential fatty acids, the information available for the composition of a lot of food items is scarce. So, in this particular case, the probability of underestimation is quite high.

Response (Revision 2): Thanks for your comment, we added more complete explanation (line number 420-423) “. Furthermore, FFQs estimate under- or over intake of food and it is considered a low-accurate method to measure absolute nutrient values. Further, the information for the availability of dietary components such as fatty acids is not sufficient. However, FFQs can assess usual and longer-term dietary intake.”

  1. Once sex was identified as a confounding variable, groups should have been separated for posterior analysis of intervention effect – this is not clear in the text. Once separated by sex and eliminated the sequence effect, the effect of time (pre-post – with-in subjects’ effect) and the intervention (HD/AD – between groups effect) can only be considered using the GML model if there is no interaction between these factors. In order to infer a difference in the changes between interventions, it’s necessary to have an interaction effect. Maybe this is why t-test were used but the rationale for the statistical analysis is not clear neither in the methods nor the results sections. Thus, with these many doubts, I cannot make a conclusive analyse of the results or the discussion.

Response: To reassure the reviewer: It is exactly what we did inside the global GLMM model and not separately as two analyses one for men and one woman (the sample size was too low and would have affected the power of the analysis). Under these circumstances, we preferred a single global model that takes everything into account. In figure 2, we used the global GLMM model. Since we have a significant effect of sex (there are no covariates only factors in GLMM), we plotted the marginal estimated means for men and women (predicted value) for each F2-isoPs provided by the overall model (not a separated analysis and the other effects were taken into account in estimated means).

Reviewer Comment (Revision 2): I do agree with this option but why then keep comparing groups with t-test and Wilcoxon test?

Response (Revision 2): Thanks for your comment, based on reviewer comment, we decided to remove the t test and Wilcoxon results and only GLMM used in the paper, moreover we updated the table 1 and 2. And results update (Line number 203-205, 228-235, 301-303

In addition, in the discussion, the term “decreased” is very confusing.

We edited and replaced with the proper words, “lower” and “was reduced” respectively (lines 247 and 305)

The terms “decrease”, “lowered”, “reduction” is also used in the results section to compare groups, namely in lines 202 and …. These terms imply changes in time and not differences between groups in the changes in time.

Response (Revision 2): Thanks for your comment, we edited and used a proper word including, higher or lower to indicate the differences between groups in the changes in time (Line number 232- 299- 311)

  1. From figure 2, none of the changes observed from pre to post intervention were negative, so there were no decreases. Increases, if significant, were larger or smaller.

Response: Figure 2 reported changes between adequate (gray bars) vs high-dairy diet (white bar according to sex). The crossover design here looks at the treatment and pre-/post is taken into account, no matter which one was started first. The significance of contrast in the figure does not indicate which we have now corrected.

Sorry, maybe the author did not understand my question because I do not understand the answer.  What I am saying is that I cannot interpret Figure 2. The legend states it refers to changes in plasma levels, but the y axis says “plasma levels”. This should be clarified. If these are changes, changes are positive, thus the values of F2-Isoprotanes increased and did not decrease. There is a difference between increasing significantly less and decreasing. This would compromise the interpretation in the discussion. If these are absolute values, it is important to state at what moment - post intervention (no matter which started first)? – and this would come back to the base line values issue.

Response (Revision 2): Thanks for your valuable comment, we edited the legend to indicate the difference between AD and HD at the end of intervention: Figure 2 “Boxplot of estimated marginal means of difference in the plasma level of F2-IsoP (ng/ml) according to sex at the end of each phase (after high-dairy (HD) and adequate-dairy (AD) diet interventions)

  1. In addition to the points already mentioned the statistic session still needs clarification regarding the following sentences:

Lines 152-155: “Comparison of baseline parameters (anthropometric parameters, glycemic and lipid profile, F2-IsoPs levels) at the beginning of each phase of the intervention was conducted using paired t-test. Comparison of anthropometric characteristics and clinical parameters between AD and HD was performed using both, paired t-tests and Wilcoxon signed-rank tests.”

– What where the authors comparing in the first sentence and in the second sentence?

Response (Revision 2): Thanks for your comment, in this study each participant had 4 visits, further, V1 and V3 considered as before HD/AD interventions however V2 and V4 considered after HD/AD). We used t-test between V1 and V3 to confirm the before beginning of AD/HD, there was no difference in anthropometric parameters, glycemic and lipid profile, F2-IsoPs levels. Moreover, we used t-test after AD/HD to find any significant differences in parameters after intervention between AD and HD. We edited the sentence “Comparison of baseline parameters (anthropometric parameters, glycemic and lipid profile, F2-IsoPs levels) before interventions (between AD and HD) was conducted using paired t-tests and Wilcoxon signed-rank tests. Comparison of anthropometric characteristics and clinical parameters after interventions (between AD and HD) was performed using, paired t-tests and Wilcoxon signed-rank tests.” (Line number 177-182)

Minor suggestions

  1. Lines 52-55: “For instance, in a randomized-parallel-group study indicated a reduction a plasma level of 15-F2t-IsoP after intake of adequate-dairy (3.5 servings/d and ≥ 1000 mg calcium/d) compared to low-dairy (<0.5 daily servings and ≤600 mg calcium/d) 54 during 12-week of intervention in obese adults with metabolic syndrome [8]. However, a 55 6-month randomized, parallel-group intervention study demonstrated no change in in the 56 plasma level of 15-F2t-IsoP after consumption of 3-5 milk servings/d compared to the con- 57 trol group (0.5-1 serving/d) among overweight men and women [9]. Despite indirect evi- 58 dence of the potential role of dairy products on oxidative stress and glycemic parameters, 59 the association between T2D risk factors and F2-IsoPs after dairy product intake remains 60 unknown in overweight individuals with hyperinsulinemia. ><0.5 daily servings and ≤600 mg calcium/d) during 12-week of intervention in obese adults with metabolic syndrome [8]. However, a 6-month randomized, parallel-group intervention study demonstrated no change in in the plasma level of 15-F2t-IsoP after consumption of 3-5 milk servings/d compared to the control group (0.5-1 serving/d) among overweight men and women [9]. Despite indirect evidence of the potential role of dairy products on oxidative stress and glycemic parameters, the association between T2D risk factors and F2-IsoPs after dairy product intake remains unknown in overweight individuals with hyperinsulinemia.”

Consider replacing with: For instance, a randomized-parallel-group study indicated a reduction in plasma level of 15-F2t-IsoP after intake of adequate-dairy (3.5 servings/d and ≥ 1000 mg calcium/d) compared to low-dairy (<0.5 daily servings and ≤600 mg calcium/d) 54 during 12-week of intervention in obese adults with metabolic syndrome [8]. However, a 55 6-month randomized, parallel-group intervention study demonstrated no change in in the 56 plasma level of 15-F2t-IsoP after consumption of 3-5 milk servings/d compared to the con- 57 trol group (0.5-1 serving/d) among overweight men and women [9]. Despite indirect evi- 58 dence of the potential role of dairy products on oxidative stress and glycemic parameters, 59 the association between T2D risk factors and F2-IsoPs after dairy product intake remains 60 unknown in overweight individuals with hyperinsulinemia. ><0.5 daily servings and ≤600 mg calcium/d) during 12-week of intervention in obese adults with metabolic syndrome [8]. However, a 6-month randomized, parallel-group intervention study demonstrated no change in the plasma level of 15-F2t-IsoP after consumption of 3-5 milk servings/d compared to the control group (0.5-1 serving/d) among overweight men and women [9]. Despite indirect evidence of the potential role of dairy products on oxidative stress and glycemic parameters, the association between T2D risk factors and F2-IsoPs after dairy product intake remains unknown in overweight individuals with hyperinsulinemia.

Response (Revision 2): Thanks for your comment and edition, We replaced with the sentences with your edition (Line number 53-61)

  1. Line 166: “(…) a p-value of less than 0.1 was considered a tendency.” Consider replacing with: a p-value of between 0.05 and 0.1 was considered a tendency

Response (Revision 2): Thanks for the edition, we replaced the related sentence (Line number 192-193)

  1. Line 178: “(…)) after AD and HD consumption (…)” consider replacing with: after AD or HD consumption

Response (Revision 2): Thanks for the edition, we delete the t-test and Wilcoxon results and add the results of the new analysis, therefore this sentence is not existing anymore,
